# Research on Decision Optimization Model of Microgrid Participating in Spot Market Transaction

**Jun Dong, Yuanyuan Wang \*, Xihao Dou, Zhengpeng Chen, Yaoyu Zhang and Yao Liu**

Department of Economic Management, North China Electric Power University, Beijing 102206, China; dongjun@ncepu.edu.cn (J.D.); xijicc@126.com (X.D.); 120192206979@ncepu.edu.cn (Z.C.); 120192206012@ncepu.edu.cn (Y.Z.); 120192206101@ncepu.edu.cn (Y.L.)
\* Correspondence: 120192206002@ncepu.edu.cn

**Abstract:** The development of electricity spot trading provides an opportunity for microgrids to participate in the spot market transaction, which is of great significance to the research of microgrids participating in the electricity spot market. Under the background of spot market construction, this paper takes the microgrid including wind power, photovoltaic (PV), gas turbine, battery storage, and demand response as the research object, uses the stochastic optimization method to deal with the uncertainty of wind and PV power, and constructs a decision optimization model with the goal of maximizing the expected revenue of microgrids in the spot market. Through the case study, the optimal bidding electricity of microgrid operators in the spot market is obtained, and the revenue is USD 923.07. Then, this paper further investigates the effects of demand response, meteorological factors, market price coefficients, and cost coefficients on the expected revenue of microgrids. The results demonstrate that the demand response adopted in this paper has better social–economic benefits, which can reduce the peak load while ensuring the reliability of the microgrid, and the optimization model also ensure profits while extreme weather and related economic coefficients change, providing a set of scientific quantitative analysis tools for microgrids to trade electricity in the spot market.

**Keywords:** microgrid; renewable energy; decision model; stochastic optimization; spot market

## 1. Introduction

### 1.1. Background

With the increasing of greenhouse gas emissions and the worsening of global environmental problems, as well as the commitments made by countries to carbon peaks and carbon neutrality, renewable energy power generation has gradually attracted wide attention from various countries. However, renewable energy such as wind and solar energy is greatly affected by the environment, and the uncertainty and intermittence of its power generation has brought huge challenges to the power system's safe operation [1]. Renewable energy mainly improves the consumption capacity through the construction of market mechanisms. However, renewable energy is difficult to directly participate in the market transaction due to its small scale and high volatility. As an important carrier of renewable energy, the microgrid has increasingly become an important part of the design of power market mechanisms. By combining renewable and non-renewable distributed energy resources, battery storage, loads, and demand responses, microgrids can facilitate the consumption of renewable energy with unpredictable and highly intermittent characteristics [2]. In 2015, the State Council has issued a guideline named "Several Opinions of the Central Committee of the Communist Party of China and the State Council on Further Deepening the Reform of the Electricity System" (ZF [2015] No. 9), marking the official launch of a new round of power market reform in China. In the new round of power market reform, the government has clearly put forward the requirements of opening the electricity sales side, enriching the types of market entities, and allowing microgrids to participate in

the power market transactions as a market entity [3]. With the continuous deepening of power system reform, the spot market has become the core and focus of China's power market construction. From the perspective of market participants, microgrids are new entrants in the deregulated electricity market, and they face various challenges and opportunities in the spot market [4]. The construction of the spot market provides a strong market mechanism guarantee for the consumption of renewable energy such as wind and PV power in the microgrid. However, under the influence of uncertain factors such as market price, renewable energy power output, and electric load, the economic dispatch of microgrids in the spot market has become a challenge. Therefore, the research on the decision-making optimization model of microgrids' participation in the spot market has practical needs and important significance.

*1.2. Literature Review*

At present, the research on microgrids mainly includes two main aspects: system optimization operation and transaction decision-making. Among them, there is relatively more research on optimal operation, mainly focusing on system operation economy, system operation security, and renewable energy uncertainty. A multi-objective optimization model suitable for multiple microgrids has been constructed, and the operating costs and emissions of multiple microgrids were minimized based on a genetic algorithm [5]. Akhtar H et al. have designed a multi-microgrid scheduling method based on robust optimization, considering the uncertainty of renewable energy power output and load [6]. Wang L et al. have considered the uncertainty of renewable energy output. Meanwhile, the integrated dispatch method based on robust multi-objective optimization has been taken to achieve the minimum operating cost of the microgrid under the worst conditions [7]. Li W et al. have taken microgrids with multiple scenarios as their research object and transformed the multi-scenario optimization problem into a dual-objective problem of minimizing the number of scenarios and minimizing the operating cost of the microgrid, and found the optimal scheduling strategy for all users [8]. The multi-energy flow microgrid has been studied with cold, heat, electricity, gas, and others. By constructing a collaborative optimization model of cold, heat, electricity, and gas, the security and economic operation of the microgrid was realized [9–11]. A scene-based optimization algorithm has been applied to deal with the uncertainty of solar irradiance and wind speed, converting the uncertain problem into certainty to solve the problem [12]. In order to solve the uncertainty of renewable energy power output and electric vehicle travel patterns, the method of robust optimization has been used to deal with the uncertain parameters, which transformed the optimization problem into a mixed integer linear programming problem to obtain the global optimal solution [13]. Mohammadi-Ivatloo B et al. have taken microgrids including gas boilers, wind turbines, and demand responses as the research object, and used a scenario-based stochastic optimization method to model the uncertainty of wind speed, electricity price, and load [14].

The above literatures are all related to the operation optimization of microgrid, while there is relatively less research on the decision-making model of microgrid participating in spot market transactions. Compared with other market entities, the microgrid has the characteristics of more energy types and a complex system benefit source. Thus, when the microgrid participates in the spot market, it is necessary to consider the uncertainty of new energy resources, and realize the maximum benefit of the microgrid while ensuring system stability. In recent years, some academics have conducted research on this issue. Most of the literature uses stochastic optimization or robust optimization methods to deal with uncertainty. A stochastic optimization method has been used to deal with the uncertainty of wind power, and the decision optimization model has been constructed of microgrids participating in the day-ahead market [15–17]. Mehdizadeh A et al. have studied the microgrid including wind power, PV, battery storage, and gas turbines, considered the demand response factors, and used the robust optimization method to model the market price uncertainty to obtain the optimal bidding strategy for the microgrid operators in



the day-ahead market [18]. A bidding strategy model has been proposed suitable for microgrids, which used stochastic optimization methods to deal with uncertain factors such as wind power, photovoltaic power, and market prices, and improved the bidding profits of microgrids in the electricity spot market [19]. A stochastic optimization method has been used to deal with the uncertainties of wind speed and market price. In addition, the bidding strategy model has been constructed of an energy hub so as to realize the optimal bidding strategy of energy hubs [20]. A two-stage method has been adopted to determine the schedule for the combined heat and power plant, the power exchanged with the grid in the day-ahead market, and the intraday optimization has been carried out according to the day-ahead optimization results and the observed values of the uncertain parameters [21]. The microgrid group containing multiple microgrids has been taken as the research object and has combined the multi-time scale optimization method with the multiplier alternating method to manage the energy of the day-ahead, intraday, and real-time markets, which eliminates the impact of uncertainty on the economic operation of microgrids to the greatest extent [22]. A stochastic optimization method has been used to deal with the uncertainty of load and renewable energy output, and a bidding strategy model for microgrids containing renewable energy has been proposed to participate in the day-ahead market, so as to maximize the microgrid's profit [23]. Aiming to jointly optimize the operation of the day-ahead market and the real-time market, the forecasted new energy output has been modeled based on mixed integer linear programming (MILP) in the day-ahead market, and the robust optimization has been used to model the uncertainty of renewable energy power in the real-time market [24]. For the two-stage optimization problem of the microgrid, most literatures use two objective functions [22,24], and some literatures use one objective function to maximize the profit of the microgrid [19–21]. When two objective functions are used, the global optimization may not be achieved by simultaneously optimizing the microgrid revenue and unit commitment. Therefore, this paper uses one objective function to maximize the revenue of microgrid in the spot market.

When stochastic optimization is applied to the processing of uncertainty in the microgrid, the significant disadvantage is that when the number of scenarios increases, the calculation requirement is high [25]. Robust optimization can only consider the best result in the worst case, which will make the result too conservative [6]. Some literatures decompose the bidding strategy of the microgrid into a two-step optimization problem. Firstly, the scenario generation and reduction of uncertain variables are completed, and then an optimization algorithm is used to determine the bidding decision and the optimal power output of each unit. A model for aggregators has been constructed with flexible power output units to participate in the spot market. Based on stochastic mixed integer linear programming, bidding decisions have been made in the day-ahead market and scheduling has been carried out in the real-time market to minimize the total transaction cost in the spot market [26]. A Monte Carlo simulation has been used to generate scenarios with uncertain variables, and a stochastic optimization method was used to maximize the profit of the bidding scheme [27]. An D et al. have constructed a stochastic double auction bidding model, and the uncertain scenarios of renewable energy power and load have been generated by Monte Carlo simulation, minimizing the operation cost of the microgrid by stochastic optimization method [28]. A random bidding strategy model has been proposed for microgrid operators in the day-ahead market. Firstly, Latin hypercube sampling has been used to generate multiple sets of scenarios, then a backward scenario reduction method has been used for scenario reduction, and finally, a stochastic optimization algorithm has been used to deal with the uncertainties of wind power and load, so as to determine the optimal bidding strategy of microgrid operators [29]. A bidding strategy model has been proposed for microgrid operators in the day-ahead market and the real-time market. Latin hypercube sampling has been used to generate scenarios of wind speed, light intensity, and load, and backward scenario reduction technology has been used to reduce the scenario [30]. Compared with the Monte Carlo simulation, the

samples generated by the Latin hypercube sampling can reflect the shape of distribution more accurately [29].

In conclusion, under the background of spot market reform, the research on microgrids with renewable energy participating in market transactions has made some progress, but there are still some problems with the research. In the above literature on the participation of microgrids in market decision-making, some do not consider the participation of the microgrid in the real-time market, some do not consider the volatility of renewable energy output power, some do not consider the uncertainty of the market price, and some only use stochastic optimization or robust optimization to solve the problems. In order to avoid the limitations of the above research, this paper uses two-step optimization to construct the decision-making model of microgrids in the spot market. Latin hypercube sampling is utilized to generate scenarios, which can accurately reflect the distribution of renewable energy, and the synchronous back generation reduction method is adopted to reduce scenarios that can reduce the amount of calculation. Considering the demand response factors, the decision-making model of microgrids participating in spot market transactions composed of wind power, PV, gas turbines, and battery storage is constructed to optimize its trading decision-making scheme. The model takes into account various constraints and microgrid components, which is more practical and provides scientific quantitative analysis tools for microgrids. In order to achieve more targeted research, this paper uses the following two hypotheses: (1) The microgrid participates in market transactions as price receivers; (2) Spot market transactions only include electricity as a subject matter.

The main contributions of this paper include the following:

1. The Latin hypercube sampling method is used to discretize the wind and PV power into multiple scenarios. This sampling method can generate more uniformly distributed sample points, which is more efficient than random sampling. In addition, to decrease the computation time, the most representative expected scenario is selected by the synchronous back generation reduction method.
2. The two-level PBDR was implemented in the day-ahead market. First, the overall load curve of the microgrid was optimized, and then, for the more flexible residential load, it can respond to changes in the power output of renewable energy, thereby maximizing the consumption of renewable energy. In the real-time market, IBDR is used to change users' electrical power consumption, which not only enables users to get a certain amount of financial compensation, but also enables microgrids to obtain part of the "negative output" at a lower electricity price.
3. The proposed stochastic bidding strategy for microgrid participation in spot market transactions, which considers the uncertainty of renewable output power, load, and market price, can ensure the maximum expected revenue of the microgrid in various scenarios.
4. The sensitivity analysis method is used to analyze the impact of each cost coefficient and market price coefficient on microgrid revenue. On the one hand, it verifies the stability of the optimization model. At the same time, it also provides a reference range for the setting of the market price coefficient.
5. The profit situation of microgrid in extreme weather is analyzed, and the microgrid can remain profitable in extreme weather, which shows that the model is stable and less affected by extreme weather.

The remainder of this paper is organized as follows. Section 2 introduces the mathematical modeling. Section 3 presents the decision optimization model of microgrid operators. Section 4 describes the numerical simulations to demonstrate the effectiveness of the proposed approach. And the abbreviations and acronyms are in Table A1.

## 2. Strategy and Modeling

In this section, the process of microgrids participating in spot market transactions is described. The mathematical model of each unit in a microgrid is given, as well as the model of uncertainty and the electricity market.

### 2.1. Spot Market Trading Strategy

The spot electricity market is also referenced as short-term electricity market. Generally, the spot market is divided into the day-ahead market and the real-time market according to time scale. In the Nordic electricity market, there is also an intraday market between the day-ahead market and the real-time market. Due to the small trading scale of intraday market in the entire spot market, it is often ignored [31]. This paper considers day-ahead market and real-time market and proposes an energy management strategy suitable for microgrids with a high renewable energy penetration rate so as to maximize the expected revenue of the microgrids.

In the day-ahead market, microgrid operators take the typical daily load curve as the actual load curve, determine and notify the user of the hourly electricity price of the next day in advance, implement the PBDR to shift peak load, and obtain new load curves. Based on the load after the implementation of the PBDR and the predicted wind and PV power, the microgrid operators call gas turbines to balance the deviation and determine the amount of bidding electrical power in the day-ahead market and its decision-making results can be used in the real-time market.

In the real-time market, due to errors in the short-term forecast of the wind and PV power, there is a deviation between the actual transaction of electrical power and the bidding on electrical power. When the forecasted wind and PV output power is greater than the actual wind and PV output power, the microgrid operators need to meet the load demand by dispatching battery storage and demand response, or they can purchase electrical power at a high price from the real-time market. When the forecasted wind and PV output is more than the actual wind and PV power, the microgrid operators can sell this part of the electrical power at a low price to the real-time market and balance the internal power of the microgrid. The spot market trading strategy is shown in Figure 1.

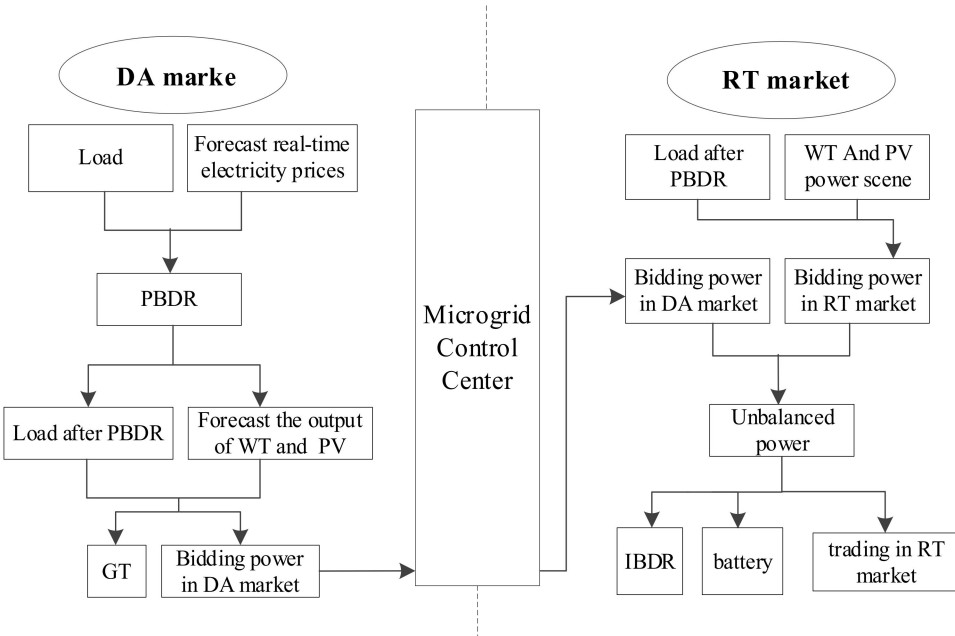

**Figure 1.** Spot market trading strategy (DA: day-ahead, RT: real-time, PV: Photovoltaic, WT: wind turbine, GT: gas turbine, IBDR: incentive-based demand response, PBDR: price-based demand response).

### 2.2. Mathematical Model of Microgrid

#### 2.2.1. Photovoltaic

Solar energy is one of the most promising renewable energy sources. At present, the utilization of solar energy mainly includes PV power generation and photothermal power generation. The scale of photothermal power generation is relatively large, and it is mostly used for centralized power generation, while PV power generation can be

applied in small-scale and distributed generation. In this paper, PV power generation is applied. PV power is closely related to the solar irradiance. Ref. [32] points out that the solar irradiance can be expressed by beta distribution function, and its probability density function is as follows:

$$f(G(t)) = \frac{\Gamma(\alpha_{sp} + \beta_{sp})}{\Gamma(\alpha_{sp})\Gamma(\beta_{sp})} \cdot \left(\frac{G(t)}{G_{\max}}\right)^{\alpha_{sp}-1} \cdot \left(1 - \frac{G(t)}{G_{\max}}\right)^{\beta_{sp}-1} \tag{1}$$

In Equation (1), $\Gamma(\bullet)$ is a gamma function that represents the solar irradiance in period $t$, $G_{\max}$ represents the maximum solar irradiance in this period in kW/m$^2$, and $0 \leq G(t) \leq G_{\max}$, $\alpha_{sp}$, and $\beta_{sp}$ are the shape parameters of beta distribution, which are determined by $\sigma_{pv}$ (standard deviation) and $\mu_{pv}$ (expected value) of the solar irradiance during this period and can be calculated based on the historical data of local solar irradiance according to Equation (2).

$$\begin{cases} \alpha_{sp} = \mu_{pv}\left[\frac{\mu_{pv}(1-\mu_{pv})}{\sigma_{pv}^2} - 1\right] \\ \beta_{sp} = (1 - \mu_{pv})\left[\frac{\mu_{pv}(1-\mu_{pv})}{\sigma_{pv}^2} - 1\right] \end{cases} \tag{2}$$

Finally, PV power is calculated according to Equation (3).

$$P_{pv}^t = P_{STC} \times \frac{G_{in}}{G_{STC}} \times \left[1 + \omega_{pv}(T_c - T_{STC})\right] \tag{3}$$

where $P_{STC}$ (kW), $G_{STC}$ (kW/m$^2$), and $T_{STC}$ (°C) indicate the rated power, solar irradiance, and PV cell temperature, respectively, of PV panels under standard test conditions. $P_{pv}^t$ represents the actual output power, in kW. $G_{in}$ refers to the actual solar irradiance. $\omega_{pv}$ is the power temperature coefficient, generally taken as $-0.47$. $T_c$ is the surface temperature of the PV panel, in °C.

### 2.2.2. Wind Turbine

At present, wind power generation has become the most mature renewable energy with the most viable development value. The wind power industry has also become one of the fastest growing and highest potential industries in the world. The principle of wind power generation is to convert wind energy into electric energy. According to its power generation principle, wind power is closely related to the wind speed. Ref. [33] points out that the actual wind speed distribution conforms to the Weibull distribution, and the probability density function of wind speed described by the Weibull distribution is as follows:

$$f(v) = \frac{a}{b} \cdot \left(\frac{v}{b}\right)^{a-1} \cdot \exp\left[-\left(\frac{v}{b}\right)^a\right] \tag{4}$$

In Equation (4), $v$ is the actual wind speed, m/s, $a$, and $b$ (m/s) represent the shape parameters and scale parameters of the wind speed curve, respectively, which can be obtained by calculating the $\mu$ (expected value) and $\delta$ (standard deviation) of historical wind speed data. The specific calculation is shown in Equation (5).

$$a = \left(\frac{\delta}{\mu}\right)^{-1.086}, b = \frac{\mu}{\Gamma(1 + \frac{1}{a})} \tag{5}$$

The relationship between wind power and wind speed satisfies the wind turbine power characteristic curve, as shown in Figure 2.

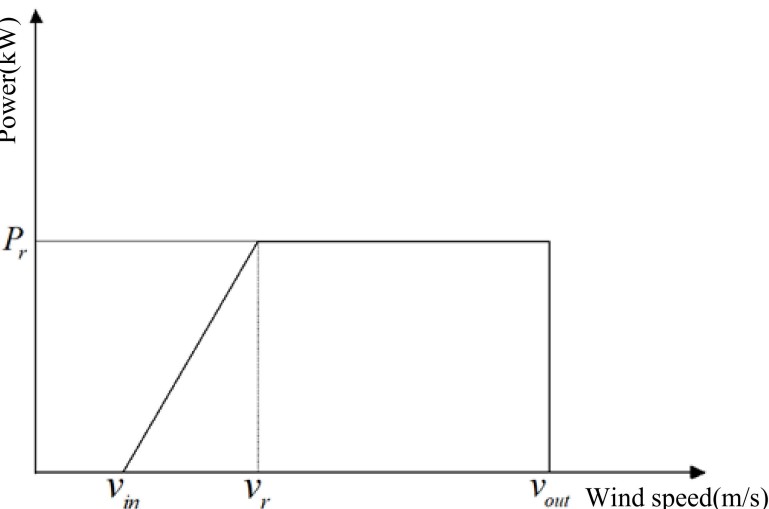

**Figure 2.** Relationship between wind speed and wind power.

According to Equations (4) and (5), the power of wind turbines can be further calculated as follows:

$$P_{wind}^t \begin{cases} 0 & ,0 \le v < v_{in} \\ \frac{P_r}{v_r^3 - v_{in}^3} v^3 - \frac{v_r^3}{v_{in}^3 - v_{in}^3} P_r & , v_{in} \le v < v_r \\ P_r & , v_r \le v \le v_{out} \\ 0 & , v > v_{out} \end{cases} \tag{6}$$

In Equation (6), $P_{wind}^t$ is the output power of the wind turbine in period $t$, kW, $P_r$ is the rated power of the wind turbine, kW, and $v_{in}$, $v_{out}$, and $v_r$ are the cut in wind speed, cut out wind speed, and rated wind speed of the wind turbine in m/s, respectively.

### 2.2.3. Gas Turbine

The gas turbine has the characteristics of fast climbing speed and low start-up and shut-down cost, which can ensure the stable operation of the microgrid in periods of low solar irradiance and lean wind and provide reserve service for the whole microgrid when the wind and PV power is insufficient. The power of gas turbines is affected by the generation efficiency, and the power model is presented in Equation (7):

$$P_{CGT}^t = W_{CGT}^t \cdot \eta_{CGT} \cdot Q_{gas} \cdot \kappa_{gas} \tag{7}$$

where $P_{CGT}^t$ is the output power of the gas turbine, kW, $W_{CGT}^t$ is the gas consumption of the gas turbine in period $t$, m$^3$, $\eta_{CGT}$ is the power generation efficiency of the gas turbine in period $t$,%, $Q_{gas}$ is the calorific value of natural gas, taking 36 MJ/m$^3$ [34], and $\kappa_{gas}$ is the conversion coefficient, 0.278 kWh/MJ.

### 2.2.4. Battery Storage

The microgrid contains a large number of distributed PV and wind turbines, which have great volatility. The installation of battery storage can ensure the stable operation of the microgrid system. In addition, battery storage has a good peak-shaving and valley-filling effect, which can store electricity when the price is low and sell electricity when the price is high, thereby improving the income of the microgrid. Therefore, considering both stability and economy, the microgrid in this paper includes battery storage. The relationship between the state of charge and the power of charging and discharging can be expressed as follows:

$$SOC^t = SOC^{t-1} + \frac{P_{ch}\eta_{ch}}{E_b} \cdot \Delta t \tag{8}$$

$$SOC^t = SOC^{t-1} + \frac{P_{dis}}{\eta_{dis}E_b} \cdot \Delta t \tag{9}$$

In Equations (8) and (9), $SOC^t$ and $SOC^{t-1}$ represent the state of charge of battery storage in period $t$ and in period $t-1$ as percentages, respectively, and $\Delta t$ represents an operating period. $P_{ch}$ and $P_{dis}$ represent the charging and discharging power of the battery storage in kW, respectively. $\eta_{ch}$ and $\eta_{dis}$ represent the charging and discharging efficiency, respectively, which is generally less than 1, and $E_b$ is the rated capacity of battery storage in kWh.

2.2.5. Demand Response

With the continuous increase of renewable energy penetration, the operation of the microgrid has been seriously affected, which may make the microgrid face the risk of abandoning wind and PV or load shedding, and demand response (DR) resources participating in the operation of microgrid is an effective way to solve this problem. DR includes the price-based demand response (PBDR) and the incentive-based demand response (IBDR). The response time of the PBDR is slow, but the economy is good, and it has a fine effect of peak-shaving and valley-filling. The IBDR has a fast response time but poor economy, which can effectively promote renewable energy consumption. According to the respective characteristics of the PBDR and the IBDR, this paper uses the PBDR in the day-ahead market to reduce the load in peak period or transfer it to low period and adopts the IBDR in the real-time market to maximize the utilization of user-side resources and promote renewable energy consumption. The adoption of the PBDR and the IBDR in the spot market considering the multi-scenarios of wind and PV power can function to better the role of the demand response, realize the stable operation of the power system, and meet the needs of market-oriented balance mechanism.

(1)    Day-ahead load dispatch based on the two-level PBDR

In the PBDR model, according to the consumer psychology principle, the reasonable "real-time electricity price" can be formulated in the day-ahead market to influence the users' electrical power consumption behavior. "Real-time electricity price" is the electricity price of the next day, which is determined in day-ahead market and notified to the user in advance. Generally speaking, the lower the electricity price, the higher the power consumption is of users, and vice versa. Based on the modeling method of stepped elastic load curve, this paper designs 10 electricity price gears. The electricity price of each gear and the response rate of the user load are shown in Table 1 [35].

**Table 1.** The price of electricity and load response rate.

| Gear | Electricity Price/(USD/kWh) | Load Response Rate/(%) |
|------|------|------|
| 1 | <0.051 | 107.9 |
| 2 | 0.051~0.059 | 104.8 |
| 3 | 0.059~0.066 | 102.3 |
| 4 | 0.051~0.059 | 100.0 |
| 5 | 0.059~0.066 | 98.0 |
| 6 | 0.066~0.073 | 96.2 |
| 7 | 0.073~0.080 | 94.6 |
| 8 | 0.080~0.087 | 93.1 |
| 9 | 0.087~0.094 | 91.8 |
| 10 | >0.094 | 90.5 |

In this paper, it is assumed that the users arrange the electrical power consumption plan in advance according to the "real-time electricity price" and determine the amount of load transfer or reduction. The mathematical model of PBDR is illustrated in Equations (10)–(14):

$$P_L^t = P_s^t \sum_{d \in D} \kappa_{dt} \sigma_{dt} \tag{10}$$

$$\sum_{d \in D} \kappa_{dt} = 1, \kappa_{dt} \in \{0, 1\} \tag{11}$$

Equation (10) represents the load value in period $t$ after the implementation of the first level PBDR, and Equation (11) indicates that the electricity price at each period can only be in one gear. Where $P_s^t$ is the total load forecast value in period $t$ in kW, $P_L^t$ is the load value after participating in the first level PBDR in kW. $\kappa_{dt}$ is a binary variable, indicating the identification of the electricity price level, and $\sigma_{dt}$ is the load response rate under the gear $d$ in period $t$.

The above is the first level PBDR for all users of the microgrid. Microgrid users can be divided into industrial, commercial, and residential users. For the residents, their electrical power consumption behavior is more flexible than that of industrial and commercial users. In order to maximize renewable energy consumption in the day-ahead market, the implementation of the second level PBDR to residential users is continued so that their load can respond to changes in wind and solar output power.

$$P_{WT,DA}^t + P_{PV,DA}^t - P_r^t = D_t \tag{12}$$

$$\triangle P_t^r = \begin{cases} 0 & , D_t < a \\ 10\% \times (D_t - a) & , D_t \geq a \end{cases} \tag{13}$$

$$P_{PB}^t = P_L^t - \triangle P_t^r \tag{14}$$

Equation (13) shows the variation of residential load value with wind and solar power output where $P_{WT,DA}^t$ is the wind power output predicted in the day-ahead market in kW, $P_{PV,DA}^t$ is the PV power output predicted by the day-ahead market in kW, $P_r^t$ (kW) is the residential load, and $D_t$ (kW) is the difference between the renewable energy power output and the residential load in the day-ahead market. $\triangle P_t^r$ is the change of residential load, and $P_{PB}^t$ is the microgrid load after the implementation of the two-level PBDR in kW. $a$ is the reference value of the difference between renewable energy power output and residential load.

(2)    Real-time load dispatch based on IBDR

In IBDR model, through signing contracts with large-load users and giving them certain economic compensation according to the contract content, users can adjust load demand, provide "negative output" for the microgrid, balance forecast errors, and improve the stability of the microgrid. The mathematical model of IBDR is shown in Equations (15) and (16).

$$0 \leq P_{IB}^t \leq \varsigma L_{PB}^t \tag{15}$$

$$-\Delta P_{IB}^{max} \leq P_{IB}^t - P_{IB}^{t-1} \leq \Delta P_{IB}^{max} \tag{16}$$

Equation (15) represents the power constraint of demand response in period $t$, and Equation (16) is the ramp rate limits of the demand response in adjacent periods where $P_{IB}^t$ is the load reduced by IBDR in period $t$ in kW, $\varsigma$ is the maximum power coefficient as a percentage, and $\Delta P_{IB}^{max}$ is the maximum ramp rate of the IBDR in kW.

### 2.2.6. Uncertain Sets of Renewable Power, Load, and Electricity Prices

There are several uncertain factors in this paper, namely, the electricity price uncertainty, the user load uncertainty, and the wind and PV power output uncertainty. Aiming at the uncertainty of electricity price, this paper uses the fuzzy clustering method to generate a set of electricity price scenarios of the day-ahead and real-time markets according to the historical electricity price data. The typical daily load curve is used as the user load scenario. Latin hypercube sampling and the synchronous back substitution reduction method are used to generate and reduce the wind and PV power scenarios to reduce the impact of uncertainty. Then, $S_{i,m}^w$ ($m$ wind speed scenes) and $S_{j,n}^{pv}$ ($n$ solar irradiance scenes) are obtained and the total number of scenes is $S$, which is equal to $m \cdot n$. The mathematical

model of the Latin hypercube sampling and the synchronous back substitution reduction method is as follows.

(1)  Latin hypercube sampling

Due to the randomness of wind and PV power, this paper introduces Latin hypercube sampling to reduce the influence of randomness on the experimental results. Latin Hypercube Sampling (LHS) is a set sampling method that guarantees full coverage of the variable range through the stratified marginal distribution to the greatest extent [29]. This sampling method can generate more uniformly distributed sample points, which is more efficient than random sampling, and is suitable for the reliability analysis in the power system. In this paper, Latin hypercube sampling is used to generate wind and PV power scenarios.

The steps of generating samples $N$ by Latin hypercube sampling are as follows:

Step 1: Determine the sample size $N$ and vector dimension $k$.

Suppose a hypercube: $x_1, x_2, x_3, \cdots, x_k$ is $k$ random variables, that is, the dimension of variables is $k$, and the cumulative probability distribution function of each variable is $F_{x_i} = f(x_i)$ and $i = 1, 2, \cdots, k$. For any random variable, $x_i$, divide the vertical axis of the cumulative probability distribution curve of $x_i$ into $N$ intervals with equal probability, the length of each interval is $1/N$, and for any value of $x$ in the interval, there are: $P(x_{ia} < x < x_{i(a+1)}) = 1/N$, where $x_{ia}$ is the $a$-th sampling value of random variable $x_i$.

Step 2: The cumulative probability distribution function corresponding to $x_{ia}$ is $F_{x_i}(a) = (1/N)r_u + (a-1)/N$, where $r_u \sim N(0,1)$ obeys a uniform distribution. The $a$-th sampling value $x_{ia}$ of $x_i$ can be obtained by calculating the inverse function of the cumulative distribution function.

Step 3: After sampling, the sampling values of $k$ random variables are arranged into a column of the matrix to form a sampling matrix $L$ of $N$ times $k$.

(2)  Synchronous back substitution reduction method

If only the above Latin hypercube sampling method is used, the sample size will be too large and the calculation efficiency will be reduced. Therefore, the samples need to be decreased [36]. In order to make the scene set obtained after scene reduction consistent with the result obtained by the original scene set, this paper uses the synchronous back generation reduction method to reduce the scene set and merge similar scenes.

Step 1: First, suppose there are $N$ different scenes $A_p$ ($p = \{1, 2 \cdots N\}$) in the initial scene set, and the probability of each scene is $\pi_P$. Obviously, the probability of each scene generated by Latin hypercube sampling is $1/N$, and $D_{p,p'}$ is defined as the distance between scenes $p$ and $p'$. $T$ is defined as the set of scenes that need to be deleted, which is empty before reduction, and the final number of scenes is $N'$. The distance between any two scenes in set $N$ is $D_{p,p'} = D(A_p, A_{p'}) = \sqrt{\sum_{i=1}^{k} \left(a_i^p - a_i^{p'}\right)^2}$, $k$ represents the dimension of each scene, and $a_i^p$ represents the value of the $i$-th dimension of scene $p$.

Step 2: Through calculation, find the scene $s$ which is closest to the scene $r$, such that $D(r,s) = \min D(r,b)$, where $r \neq b, r, b \in P$.

Step 3: The probability of scene $r$ is $\pi(r)$, and the probability distance between scene $r$ and $s$ is $PD_{r,s}$, so that $PD_{r,s} = \pi(r) \cdot D(r,s)$ and $PD_{r,s} = \min PD_{d,s}$ determines the scene $d$ to be cut.

Step 4: Change the total number of scenes to $S = S - \{d\}, T = T + \{d\}$. The probability of the scene becomes $\pi_r = \pi_r + \pi_d$, so that the sum of the probabilities of all scenes is always 1.

Step 5: Repeat steps 2, 3, and 4 until the final scene number is obtained.

### 2.2.7. Market Model

In the microgrid, the electrical power trading is conducted between the microgrid operators and the electrical power users. In order to prevent market players from speculating in the real-time market, the electrical power buying price in the real-time market is usually



higher than that in the day-ahead market, while the electrical power selling price is lower than that in the day-ahead market. Thus, because the bidding power of the microgrid participating in market transactions is limited, which is not enough to affect the market price, this paper assumes that the microgrid participates in the spot market transaction as a price-taker, and the transaction price is the unified clearing price. The market bidding strategy contains many complex economic problems, which are solved according to the electricity generation, load, and time-of-use electricity price. The detailed steps for signing the bilateral contracts are shown in [37], which are not detailed here.

## 3. Optimization Model of the Microgrid Participating in Market Transaction Decisions

In this section, the objective function and constraints are described of microgrid optimization, and the process of model solving is illustrated.

### 3.1. Objective Function

In the actual operation process of the microgrid with distributed energy, due to the uncertainty of the renewable energy power, a variety of energy equipment had to be dispatched. At the same time, because the small-scale energy of the microgrid itself does not meet the large-scale energy input and output, it needs energy transactions with the grid to ensure the safe and stable operation of the system. Therefore, the following key factors related to microgrid revenue need to be considered: uncertainty of wind and PV power, the coordinated operation of multiple energy equipment, and maximizing the expected revenue of microgrid.

Microgrid operators apply for bidding power in the day-ahead market, and dispatch gas turbines, battery storage, and demand response in the real-time market to make up for the unbalanced power deviations of the entire microgrid. This paper takes day-ahead and real-time transaction sequence as two stages to optimize the unit commitment, and the objective function is to maximize the total revenue of the spot market, as shown below:

$$f = max \sum_{t=1}^{T} \sum_{i=1}^{m} \rho(S_{i,m}^{w}) \sum_{j=1}^{n} \rho(S_{j,n}^{pv}) \{ I_{load}^{t} + I_{DA}^{t} + I_{RT}^{t,S_{i,m}^{w}, S_{j,n}^{pv}} - C_{CGT}^{t} - C_{Ess}^{t} - C_{IB}^{t} \} \quad (17)$$

where $I_{load}^{t}$ is the income of microgrid selling electricity to users, $I_{DA}^{t}$ and $I_{RT}^{t,S_{i,m}^{w}, S_{j,n}^{pv}}$ are the income of the microgrid in the day-ahead market and real-time markets in USD, respectively. $C_{CGT}^{t}$ represents the cost of the gas turbine, $C_{Ess}^{t}$ is the charge and discharge cost of the battery storage, and $C_{IB}^{t}$ represents the demand response cost in USD.

(1) The $I_{load}^{t}$ is calculated as:

$$I_{load}^{t} = P_{s}^{t} \cdot \varepsilon_{DA}^{t} \cdot \Delta t \quad (18)$$

The main source of income of the microgrid is the sale of electric energy to users where $\varepsilon_{DA}^{t}$ is the electricity price in the day-ahead market obtained by clustering in USD/kWh, and $\varepsilon_{DA}^{t}$ is less than the electricity purchase price of users in the market, so that users choose to buy electricity on the microgrid.

(2) The $I_{DA}^{t}$ is calculated as:

$$I_{DA}^{t} = [(1 - \mu) \cdot \varepsilon_{DA}^{t} \cdot P_{DA}^{t,sell} - (1 + \mu) \cdot \varepsilon_{DA}^{t} \cdot P_{DA}^{t,buy}] \cdot \Delta t \quad (19)$$

$$P_{DA}^{t,sell} = max\{P_{DA}^{t}, 0\} P_{DA}^{t,buy} = max\{-P_{DA}^{t}, 0\} \quad (20)$$

Equation (19) represents the income of microgrid participating in the day-ahead market. Equation (20) is used to obtain the power of purchase and sale in the day-ahead market, respectively, where $P_{DA}^{t,sell}$ refers to the sale of surplus electrical power to the market, $P_{DA}^{t,buy}$ refers to the purchase of shortages electrical power from the market in kW, and $\mu$ is the price coefficient.

(3) The $I_{RT}^{t,S_{i,m}^w, S_{j,n}^{pv}}$ is formulated as:

$$I_{RT}^{t,S_{i,m}^w, S_{j,n}^{pv}} = \rho(S_{i,m}^w)\rho(S_{j,n}^{pv})[(1-\delta) \cdot \varepsilon_{RT}^t \cdot P_{RT,sell}^t - (1+\delta) \cdot \varepsilon_{RT}^t \cdot P_{RT,buy}^t] \cdot \Delta t \quad (21)$$

$$P_{RT,sell}^t = max\{P_{RT}^{t, S_{i,m}^w, S_{j,n}^{pv}}, 0\}, \quad P_{RT,buy}^t = max\{-P_{RT}^{t, S_{i,m}^w, S_{j,n}^{pv}}, 0\} \quad (22)$$

Equation (21) represents the income in real-time market. Equation (22) is used to obtain the power of purchase and sale in the real-time market, respectively, where $P_{RT}^{t, S_{i,m}^w, S_{j,n}^{pv}}$ is the transaction power of the microgrid under different scenarios in kW, $P_{RT,sell}^t$ represents the sale of electrical power to the market in period $t$, and $P_{RT,buy}^t$ represents the purchase of electrical power from the market in kW. $\delta$ is the electricity price coefficient of purchase and sale, $\rho(S_{i,m}^w)$ and $\rho(S_{j,n}^{pv})$ are the probability of different simulated scenarios of wind and PV power, respectively, and $\varepsilon_{RT}^t$ is the electricity price in the real-time market in USD/kWh.

(4) The calculation of $C_{CGT}^t$ is given in Equations (23)–(25):

$$C_{CGT}^t = C_{CGT}^{t,ss} + C_{CGT}^{t,pg} \quad (23)$$

$$C_{CGT}^{t,ss} = g_{CGT} \cdot \left[u_{CGT}^{t+1}(1 - u_{CGT}^t) + u_{CGT}^t \cdot \left(1 - u_{CGT}^{t+1}\right)\right] \quad (24)$$

$$C_{CGT}^{t,pg} = \gamma_{CGT} \cdot P_{CGT}^t \cdot \Delta t \quad (25)$$

where $C_{CGT}^{t,ss}$ and $C_{CGT}^{t,pg}$ are the start-stop operating cost of a gas turbine in USD, $g_{CGT}$ (USD/time) and $\gamma_{CGT}$ (USD/kWh) represent the start-stop and operating cost coefficients of a gas turbine, respectively.

(5) The calculation of $C_{Ess}^t$ is given in Equation (26):

$$C_{Ess}^t = \partial_{Ess} \cdot (P_{ch}^t + P_{dis}^t) \cdot \Delta t \quad (26)$$

where $\partial_{Ess}$ is the charge and discharge cost coefficient of battery storage in USD/kWh.

(6) The calculation of $C_{IB}^t$ is given in Equation (27):

$$C_{IB}^t = \vartheta_{IB} \cdot P_{IB}^t \cdot \Delta t \quad (27)$$

where $\vartheta_{IB}$ is the cost coefficient of the IBDR in USD/kWh.

*3.2. Constraints*

(1) Power balance constraints:

$$P_{DA}^t = P_{WT,DA}^t + P_{PV,DA}^t + P_{CGT}^t - P_{PB}^t \quad (28)$$

Equation (28) represents the power balance constraint in the day-ahead market, where $P_{WT,DA}^t$ and $P_{PV,DA}^t$ represent the forecasted power of wind and PV, respectively, $P_{PB}^t$ is the load value after the implementation of the PBDR, and $P_{DA}^t$ is the bidding power of microgrid in kW.

$$P_{RT}^{t, S_{i,m}^w, S_{j,n}^{pv}} = P_{S_{i,m}^w}^t + P_{S_{j,n}^{pv}}^t + P_{CGT}^t + P_{dis}^t - P_{ch}^t + P_{IB}^t - P_{PB}^t - P_{DA}^t \quad (29)$$

Equation (29) represents the power balance constraint in the real-time market. Among them, $P_{RT}^{t, S_{i,m}^w, S_{j,n}^{pv}}$ is the transaction power of microgrid in different scenarios, $P_{S_{i,m}^w}^t$ and $P_{S_{j,n}^{pv}}^t$ represent the wind and PV power in different scenarios in kW, respectively.

(2)  Wind constraints:

$$0 \leq P_{S_{i,m}^w}^t \leq P_r \tag{30}$$

Equation (30) represents the wind power constraints, where $P_{S_{i,m}^w}^t$ represents the actual wind power in period $t$ and $P_r$ is the rated power, which is assumed to be the maximum power in kW.

(3)  PV constraint:

$$0 \leq P_{S_{j,n}^{pv}}^t \leq P_{STC} \tag{31}$$

Equation (31) is the PV's power constraints, where $P_{S_{j,n}^{pv}}^t$ represents the actual PV power in period $t$ and $P_{STC}$ is the rated power, which is presumed to be the maximum power in kW.

(4)  Gas turbine constraints:

$$\left( Q_{CGT}^{on} - R_{CGT}^{t-1,on} \right)\left( u_{CGT}^t - u_{CGT}^{t-1} \right) \geq 0 \tag{32}$$

$$\left( Q_{CGT}^{off} - R_{CGT}^{t-1,off} \right)\left( u_{CGT}^{t-1} - u_{CGT}^t \right) \geq 0 \tag{33}$$

$$u_{CGT}^t P_{CGT}^{\min} \leq P_{CGT}^t \leq u_{CGT}^t P_{CGT}^{max}, \ u_{CGT,t} \in \{0,1\} \tag{34}$$

$$\begin{cases} P_{CGT}^t - P_{CGT}^{t-1} \leq RU_{CGT} \cdot \Delta t \\ P_{CGT}^{t-1} - P_{CGT}^t \leq RD_{CGT} \cdot \Delta t \end{cases} \tag{35}$$

Equations (32) and (33) limit the start-up and shut-down time of gas turbine, and it cannot be started up and shut down simultaneously at any time. Among them, $u_{CGT}^t$ is a binary variable, which represents the operating state of the gas turbine in period $t$. When $u_{CGT}^t$ is equal to 1, the gas turbine is in the operating state, otherwise it is in the shutdown state. $Q_{CGT}^{on}$ and $Q_{CGT}^{off}$ represent the minimum start-up time and shutdown time, respectively. $R_{CGT}^{t-1,on}$ and $R_{CGT}^{t-1,off}$ respectively represent the continuous operation time and continuous shutdown time of gas turbines. Equation (34) is the power constraints, and Equation (35) is the ramp rate limits, where $RD_g$ and $RU_g$ represent the landslide rate and ramp rate in kW/h, respectively, and $\Delta t$ represents an operating period.

(5)  Battery storage constraints:

$$E_b \cdot SOC_{\min} \leq E_b^t \leq E_b \cdot SOC_{\max} \tag{36}$$

$$\omega_{ch}^t P_{ch}^{min} \leq P_{ch}^t \leq \omega_{ch}^t P_{ch}^{max} \tag{37}$$

$$\omega_{dis}^t P_{dis}^{min} \leq P_{dis}^t \leq \omega_{dis}^t P_{dis}^{max} \tag{38}$$

$$0 \leq \omega_{ch}^t + \omega_{dis}^t \leq 1, \ \omega_{ch}^t, \omega_{dis}^t \in \{0,1\} \tag{39}$$

Equation (36) expresses the capacity constraints of the battery storage. Equations (37) and (38) express the power limits of charging and discharging, respectively. Equation (39) ensures that the battery storage cannot be charged and discharged at the same time, where $SOC_{\min}$ and $SOC_{\max}$ represent the minimum and maximum state of charge, respectively. $\omega_{ch}^t$ and $\omega_{dis}^t$ are binary variables, which represent the charge and discharge state of battery storage in period $t$, respectively.

### 3.3. Model Solution

Thus far, the decision-making model is constructed. This paper develops a YALMIP-based binary mixed integer linear programming microgrid optimization model. YALMIP is a solution toolbox written in MATLAB language, which optimizes the objective function by calling a variety of commercial optimization solvers, such as LPSOLVE, CPLEX, and GRUOBI, etc. This article is programmed in the Matlab2016a environment and calls the

CPLEX college edition solver to optimize the problem. The decision model solving process is shown in Figure 3.

**Figure 3.** Integrated solution framework.

In MATLAB programming, decision variables and zero one variables related to gas turbine, battery storage, demand response, and other components are input by ">> x = sdpvar (1,24)" and >> "x = binvar (1,24)", then, the end cycle functions of gas turbine and energy storage are input, then the output power constraints and the system power balance constraints of each unit are input by ">> F = set (constraint)", and then input ">> ops = sdpsettings ('solver', 'cplex', 'verbose', 2)", call the CPLEX solver, and finally, input "result = optimize (F,f,ops)" to solve the minimization problem of the mathematical programming. Since we want to solve the problem of maximizing the expected return, we only need to add a negative sign before f.

## 4. Case Study

This section mainly presents basic data, gives the optimization results, and analyzes the impact of the demand response, meteorological factors, market price, and cost coefficients on the expected revenue of microgrid.

### 4.1. Data and Description

In this section, taking into account the time characteristics of wind power, PV, and load, as well as the volatility of electricity prices, the bidding strategy of the microgrid in spot transactions is presented. This paper selects the historical output power data of wind power, PV, and user load in a residential area in Beijing, China. The electricity price data is obtained by fuzzy clustering, and the day-ahead market and real-time market electricity price curve is demonstrated in Figure 4. A typical daily load curve is shown in Figure 5. One thousand sets of wind speed and solar irradiance scenarios are generated by Latin hypercube sampling, respectively, and then reduced by the synchronous back substitution reduction technique. Finally, 10 sets of wind speed scenarios and 5 sets of

solar irradiance scenarios with weights are obtained, and the wind and PV power are then calculated according to the formula, which constitute 50 sets of wind and PV power scenes with weights. The 10 sets of wind power and 5 sets of PV power scenarios are denoted in Figures 6 and 7, respectively.

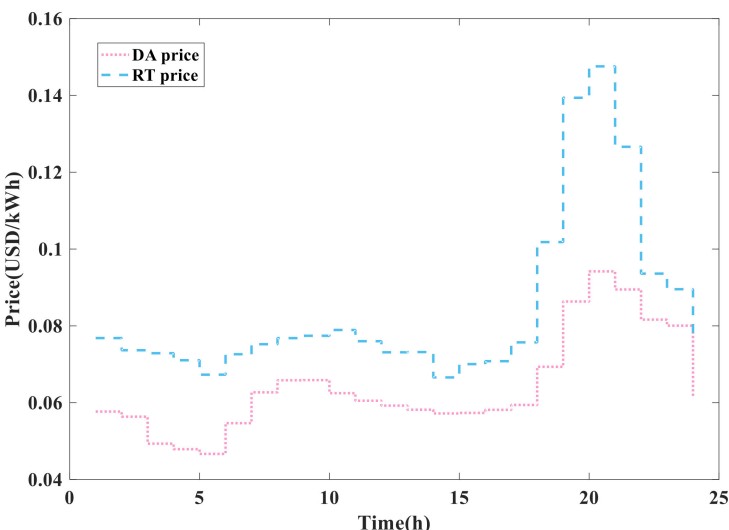

**Figure 4.** Profile of changes in the day-ahead and real-time market prices.

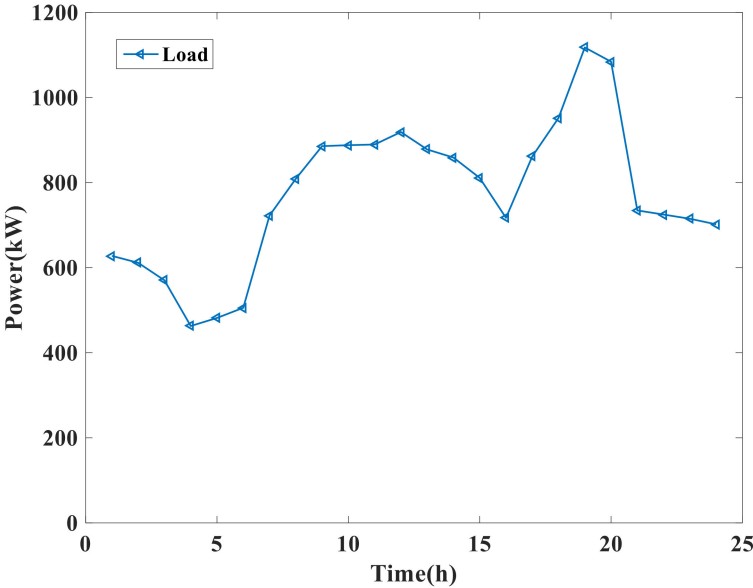

**Figure 5.** Load in a typical day.

Figure 4 denotes that the real-time market electricity price is generally higher than the day-ahead market, and both of them are in the low electricity price period from 1:00 to 6:00, and reached the peak electricity price period between 18:00 and 22:00.

As shown in Figure 5, the low period of power consumption is between 1:00 and 6:00, and the peak period of power consumption is between 17:00 and 22:00, which are generally consistent with the changing trend of the electricity price, indicating the validity of the data.

For Figure 6, the wind power shows a certain degree of uncertainty at each time, and fluctuates greatly in a short period of time. As clearly observed from Figure 7, PV power has a certain periodicity in a day. With the change of solar irradiance, PV power is in an upward trend in the morning, and downward in the afternoon, and peaks at noon. The PV power in different dates has certain similarities.

Finally, the main parameters involved in the microgrid system are shown in Tables 2 and A2–A4.

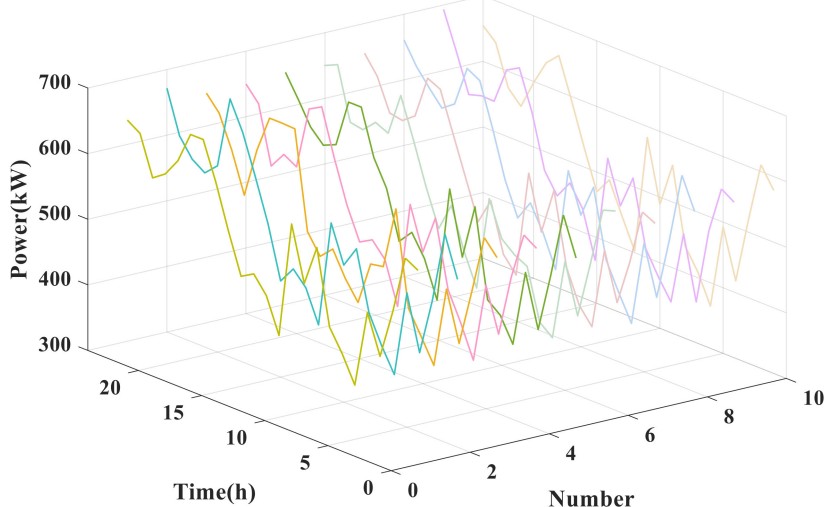

**Figure 6.** The scenes of wind power.

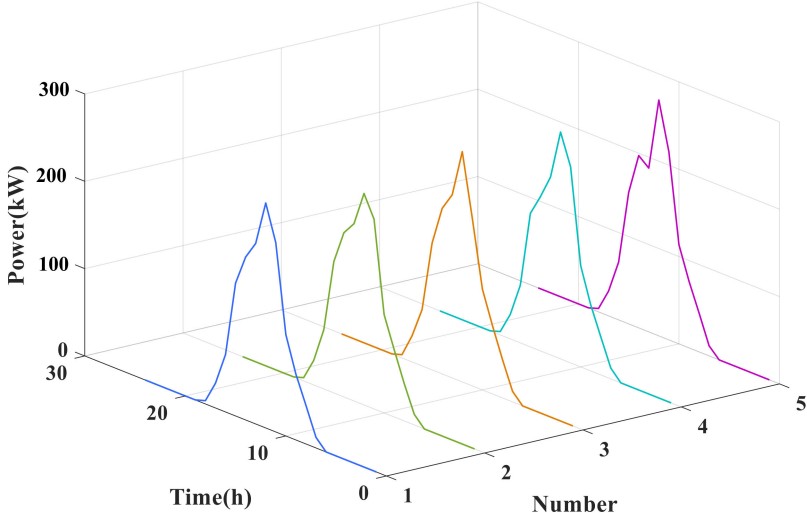

**Figure 7.** The scenes of PV power.

**Table 2.** Technical parameters of the microgrid system.

| Parameters | Value | Parameters | Value |
|---|---|---|---|
| $P_{STC}$ | 300 kW | $Q_{CGT}^{off}$ | 1 h |
| $G_{STC}$ | 1 kW/m2 | $P_{CGT}^{min}$ | 10 kW |
| $T_{STC}$ | 25 °C | $P_{CGT}^{max}$ | 100 kW |
| $P_r$ | 650 kW | $\eta_{ch}, \eta_{dis}$ | 0.95 |
| $v_{in}$ | 3 m/s | $P_{ch}^{max}$ | 15 kW |
| $v_{out}$ | 20 m/s | $P_{dis}^{max}$ | 20 kW |
| $v_r$ | 11 m/s | $E_b$ | 100 kWh |
| $\eta_{CGT}$ | 0.8 | $SOC_{min}$ | 10% |
| $RD_g, RU_g$ | 20 kW/h | $SOC_{max}$ | 90% |
| $\varsigma$ | 0.2 | $\gamma_{CGT}$ | USD0.05/kWh |
| $\varepsilon_{DA}^t$ | 0.2 | $g_{CGT}$ | USD45/time |
| $\varepsilon_{RT}^t$ | 0.6 | $\partial_{Ess}$ | USD0.10/kWh |
| $Q_{CGT}^{on}$ | 2 h | $\vartheta_{IB}$ | USD0.11/kWh |

*4.2. Results and Discussion*

4.2.1. Operation Analysis of Microgrid

Figure 8 shows the power of each component of the microgrid, including load, PV, wind turbine, gas turbine, battery, IBDR, and transaction power in the day-ahead market and the real-time market.

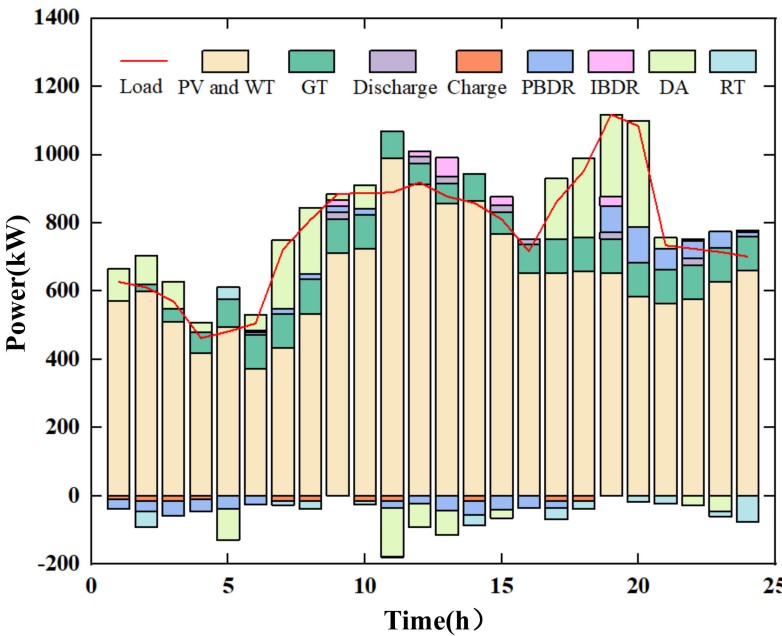

**Figure 8.** Operating status of microgrid system components.

As shown in Figure 8, since the user load is greater than the wind and solar output, the microgrid needs to predict the wind and PV output power in the day-ahead market and use lower-cost gas turbines to meet part of the load. The unsatisfied part is obtained through day-ahead market transactions. Because the cost of battery storage and demand response is higher than the electricity purchase price in the day-ahead market and the output power is relatively small, they are used to balance the volatility of wind and PV output power in the real-time market. The unbalanced part needs to be balanced by real-time market transactions.

Compared with the real-time market, the electricity selling price is higher and the buying price is lower in the day-ahead market. Thus, microgrid operators can trade more electrical power in the day-ahead market to maximize the expected revenue. Figure 8 shows that the day-ahead market transaction electrical power occupies the vast majority of the spot market, which is similar to the general transaction model in the spot market.

The power scheduling results of battery storage, gas turbines, and IBDR are denoted in Figure 8. Battery storage works in charging mode from 0:00–4:00 and 7:00–10:00, when the wind and PV power is sufficient, or electricity prices are relatively lower. Battery storage is discharging between 12:00–15:00 and 19:00–21:00, when electricity prices are comparatively higher. Therefore, the revenue of battery storage is derived from the difference between the on-peak and off-peak periods. Gas turbines provide power between 2:00 and 24:00. There is a certain difference between wind and PV output power and user load, and the cost of gas turbines is lower than the electricity purchase price in the day-ahead market and the rated power is larger, but the start and stop costs are higher, therefore, it is in the state of power generation within a day. The IBDR mainly provides "negative power" to microgrid from 12:00 to 15:00, when the cost of load reduction is lower than the electricity price in the real-time market. It can not only enable users participating in IBDR to obtain additional benefits, but also enable the microgrid operators to get part of the output power at a lower price.

From the perspective of the microgrid, the model calls gas turbine and the PBDR in the day-ahead market, and battery storage and the IBDR in the real-time market to minimize the transaction volume in the market, so that the microgrid achieves a power balance and maximizes its expected revenue, which has important reference significance for microgrids to participate in spot market transactions.

### 4.2.2. Analysis of Influencing Factors

Based on the above optimization results, this section further analyzes the impact of demand response, meteorological factors, electricity price coefficients, and cost coefficients on the expected revenue of microgrid.

(1)   Demand response

This paper introduces the PBDR and the IBDR to participate in the optimal dispatch of the microgrid. In order to further research the impact of the PBDR and the IBDR on the microgrid, four cases are compared: (i) case 1: with PBDR and IBDR; (ii) case 2: with IBDR only; (iii) case 3: with PBDR only; and (iv) case 4: without DR.

Figure 9 demonstrates the load reduction in different cases. After the application of PBDR, the power demand is increased between 12:00 and 18:00, and reduced between 19:00 and 23:00 due to changes in electricity prices. Users will reduce power consumption during periods of high electricity prices and shift their load to periods of low electricity prices. After the operation of the IBDR, users reduced power demand between 18:00 and 22:00, when the load reduction cost of the microgrid is lower than the electricity price in the real-time market. After implementing PBDR and IBDR, the peak load can be shifted to the trough period, and part of the peak load can also be decreased, which can promote the consumption of renewable energy and increase the revenue of microgrid. It can be observed that the market transactions that include both the IBDR and the PBDR have great social and economic benefits.

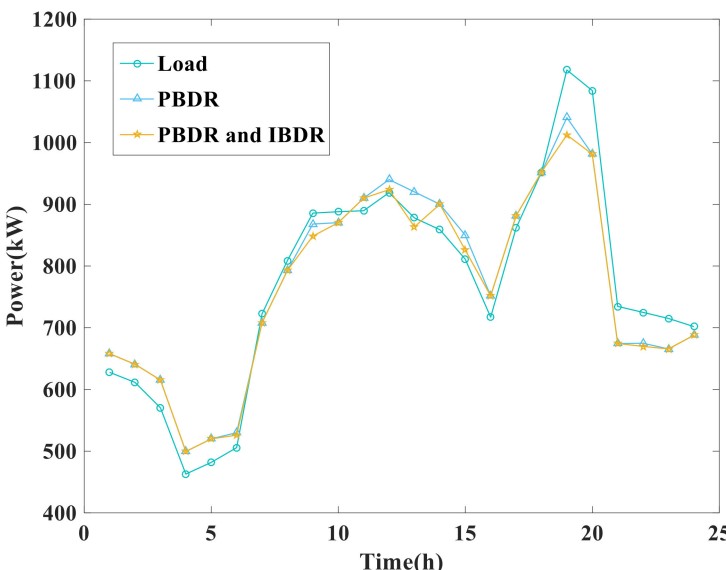

**Figure 9.** Load reduction after demand response.

The expected revenue of the microgrid in different cases is tabulated in Table 3. It can be found that the revenue in case 1 is USD 962.21, which higher than the other cases. The revenue of case 4 is reduced by 0.97% compared with case 1. In case 4, the microgrid cannot adjust the load flexibly, and can only purchase electrical power at a high price when there is a power shortage. Therefore, more revenue can be made by the intended method and the economic superiority is confirmed.

**Table 3.** Expected revenue of microgrid in different cases.

| Case | Case1 | Case2 | Case3 | Case4 |
|---|---|---|---|---|
| Expected revenue (USD) | 962.21 | 957.11 | 957.99 | 952.80 |

Therefore, the microgrid should implement the PBDR and the IBDR, which can not only shift or reduce load peaks in the day-ahead market, but also reduce the power consumption of users through certain incentives in the real-time market and improve the revenue of microgrid operators while also ensure the stable operation of the system.

(2)    Meteorological factors

Considering the rapid development of environmental issues such as pollution and climate change, there is a higher probability of extreme weather occurring more frequently, which brings huge economic losses and adverse social impacts. At this stage, wind power is developing rapidly, and the installed capacity is gradually increasing. With the level of wind power connected to the microgrid becoming higher and higher, the impact of its power output on the system is becoming ever more obvious. The stable operation of microgrid under extreme weather is of great significance to guarantee people's daily production and life, ensure the revenue of the microgrid, and promote economic and social development. As the wind speed is extremely high or low in extreme weather, and the wind speed is closely related to the wind power, in order to avoid the greater impact of wind power fluctuations on the microgrid, and to ensure power quality and safety, this section simulates the change of wind power under extreme weather to determine if the model behaves as expected.

In Figure 10, 10 sets of wind power scenes under weak wind, normal wind, and fierce wind are simulated, respectively. In addition, suppose that under weak wind and fierce wind conditions, the value of wind power fluctuates near the minimum and maximum value of the normal power, respectively.

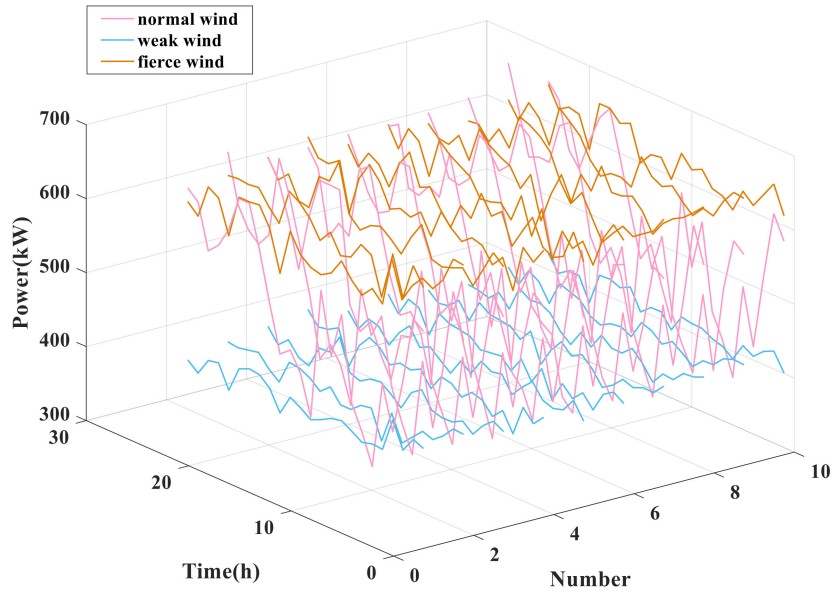

**Figure 10.** Wind power scenarios in different climatic conditions.

It can be seen in Figure 11 that different wind speeds have different effects on the expected revenue, and as the wind speed increases, the expected revenue also gradually increases. This is because the larger the wind speed, the larger the wind power output, which makes microgrid operators able to trade more electrical power in the market and then obtain more expected revenue. In addition, under weak wind conditions, the microgrid can

still make a profit, which demonstrates that the model can ensure the safe and economic operation of the microgrid and has excellent resistance to extreme weather interference.

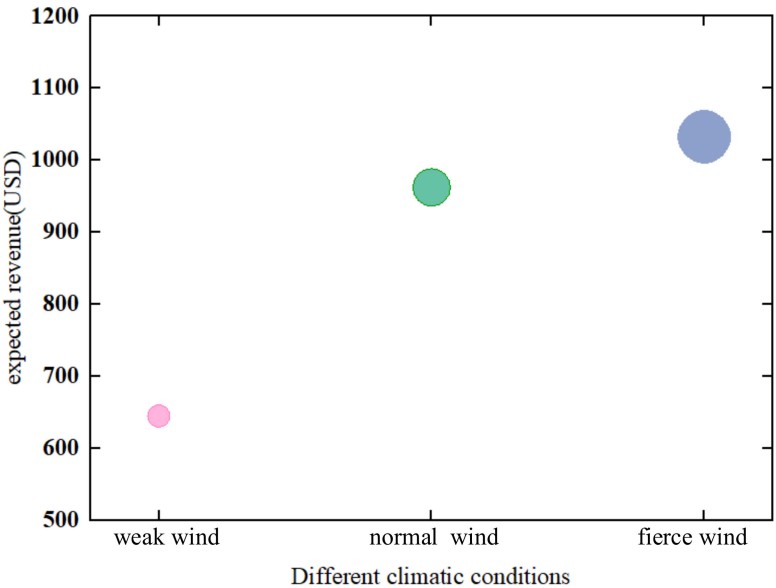

**Figure 11.** Expected revenue of microgrid under different climatic conditions.

(3)   Sensitivity analysis

Due to the randomness and uncontrollability of renewable energy power, higher requirements are put forward for the transaction decisions of the microgrid. Therefore, it is necessary to study critical operational indicators related to spot market transactions in order to promote microgrid operators to make the best decisions and maximize the expected revenue. The expected revenue of the microgrid is affected by operating income and cost. Operating income is influenced by the day-ahead market price coefficients $\mu$ and real-time market price coefficients $\delta$, and operating cost is mainly affected by the cost coefficient of each unit. To obtain the influence of various factors on the expected revenue, a sensitivity analysis was conducted based on the above research. The value of the six coefficients is changed by decreasing and increasing 10%, 20%, 30%, 40%, and 50%, respectively, based on the original data. Figure 12 gives principal factors that affect the expected revenue of the microgrid. Figures 13 and 14 denote the sensitivity analysis of the impact of the operating incomes and the operating cost on the expected revenue, respectively.

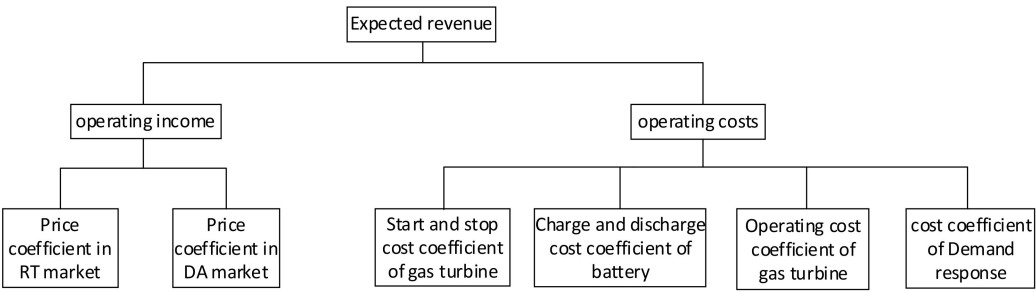

**Figure 12.** Main influencing factors of expected revenue.

As shown in Figure 13, the changes of $\mu$ (price coefficient in the DA market) has a slight effect on the result, and the fluctuations of $\delta$ (price coefficient in RT market) has a more obvious effect on the result. This is due to the fact that the real-time market electricity price is relatively higher. With the decrease of $\delta$, electricity selling prices will gradually increase to be even larger than the day-ahead market. Therefore, $\delta$ is one of the sensitive factors during the optimization model. Nevertheless, it can also be learned that the change

of $\delta$ cannot notably affect the revenue. No matter how the price coefficient fluctuates, the revenue will not decrease significantly, which can verify the effectiveness of the model to a certain degree.

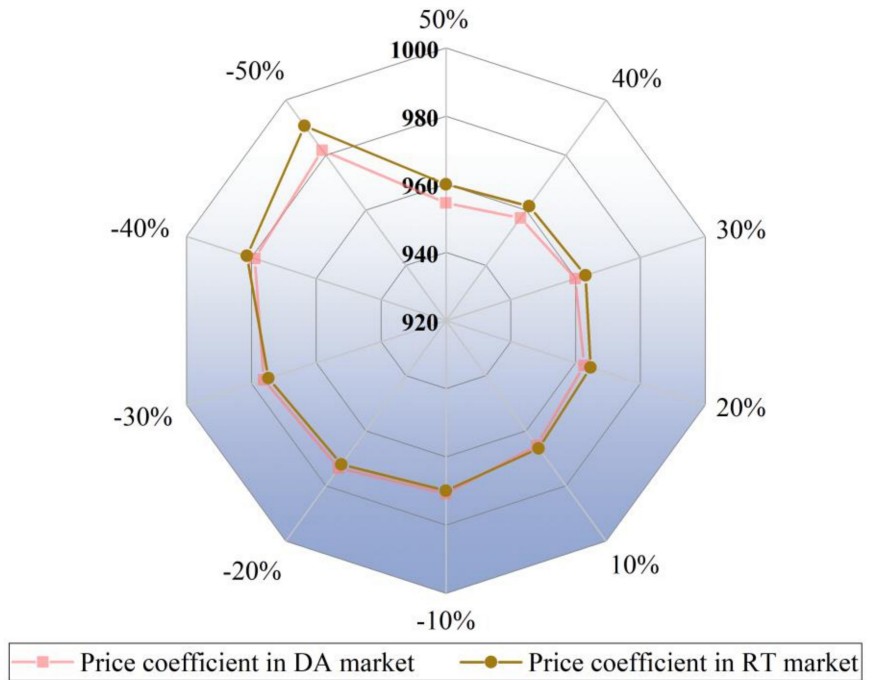

**Figure 13.** Fluctuation of electricity price coefficients.

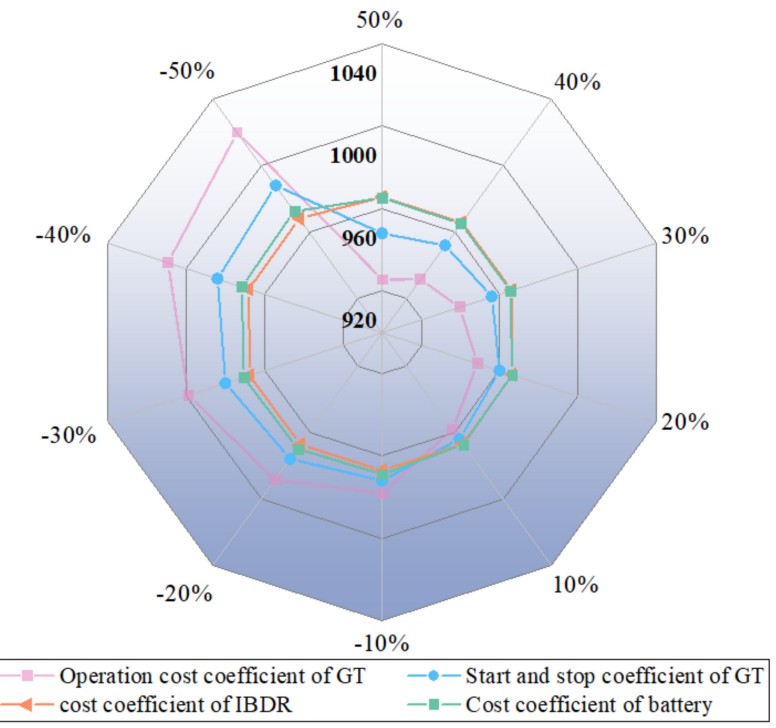

**Figure 14.** Fluctuation of cost coefficient of each unit.

With respect to the cost coefficient aspect, it can be seen from Figure 14 that no matter how $\partial_{Ess}$ (cost coefficient of battery) and $\vartheta_{IB}$ (cost coefficient of IBDR) change, the expected revenue still experiences little fluctuations. When $g_{CGT}$ (start and stop cost coefficient of GT) changes, the expected revenue fluctuates slightly. When $\gamma_{CGT}$ fluctuates, the result

fluctuates more obviously. The battery storage and IBDR have a small impact on the results due to lower power relatively. The start and stop cost of gas turbines is relatively high, but it only starts and stops once, therefore its change has a certain degree of effect on the results. The gas turbine provides the maximum output power to cover the constant load within 24 h, therefore the change in $\gamma_{CGT}$ has the most obvious impact on the result. However, no matter how each cost coefficient changes, the impact on the expected revenue of the microgrid remains within a specified range, which verifies the stability of the model to a certain extent.

Depending on the above analysis, $\delta$ and $g_{CGT}$ have the greatest impact on the expected revenue. As $\delta$ decreases, although it will increase the expected revenue of the microgrid, it will make the electricity selling price in the real-time market larger than the day-ahead market, disrupting market rules, and making the market entities speculate. Thus, it is necessary to control the change range of $\delta$ between $-10\%$ and $50\%$ to ensure the effective operation of electricity market. For gas turbines, due to the large amount of power generation, the cost has a greater impact on the results. On the one hand, the start-up and shutdown costs are high, and microgrid operators should control the start-up and shutdown of the units. On the other hand, they should strive to improve the efficiency and reduce the energy loss, which can greatly improve the income of the microgrid.

### 4.3. Operational Strategy

In this paper, we assume that the microgrid system is consistent with the interests of users, and analyzes the impact of various factors of the microgrid system on its expected return, and then maximizes its expected return in the spot market. First, it analyzes the impact of the four demand response modes on the expected return of the microgrid. When the PBDR and IBDR are implemented at the same time, it can not only maximize the expected return of the microgrid, but also reduce the peak–valley difference and ensure the stable operation of the microgrid system. Therefore, implementing both the PBDR and IBDR at the same time is the best option of the microgrid. Secondly, the impact of extreme weather on expected revenue is analyzed. When the wind power generation level is low, the microgrid can still maintain high profitability, which shows that the model has good stability and is suitable for microgrid systems in various regions. Finally, a sensitivity analysis is made on the factors affecting the expected return of the microgrid system. On the one hand, the microgrid can maintain a high expected return when various factors change. On the other hand, it provides a reference for the selection of units for the microgrid, and also provides a reference interval for the formulation of market price coefficients. In summary, the bidding strategy of the microgrid in the spot market is a complex project, which not only maximizes the expected revenue but also considers various factors within the system (e.g., unit output constraints, power balance constraints) and the external environment (e.g., weather conditions, operation behavior).

### 5. Conclusions

This paper is based on the background that the microgrid can participate in market transactions as a subject under the China's latest spot market reforms and establishes a two-step optimization model considering the uncertainties of wind power and PV power. Moreover, this paper optimizes the bidding electrical power of the microgrid in the day-ahead and real-time markets as well as the power output of the battery storage, micro gas turbine, and demand response. It also maximizes the expected revenue of the microgrid. Simulations are delivered to demonstrate the advantages of the decision-making model.

The major conclusions of this paper are as follows: (1) By constructing the decision-making model of microgrid participating in spot market, the transaction power of microgrid is obtained, which provides quantitative analysis tool for the microgrid to participate in spot market transaction decisions; (2) Through the comparative analysis of various demand responses, the PBDR and IBDR play an important role in peak cutting and valley filling, which can transfer peak load to low period and provide a partial "negative output",

promote the consumption of renewable energy, and improve the expected revenue of the microgrid; (3) In extreme weather, the microgrid can still maintain revenue, which indicates that the stability of the microgrid is good; and (4) Through the comparison and analysis of various economic coefficients, the stability of the model is verified, and the reference interval for the setting of market price coefficient is provided.

The work is of great significance for small microgrids to participate in spot market transactions, especially for the areas with high renewable energy penetration. The application of the decision-making model can be referred to, but considering that there may be different types of problems when it is extended to a large-scale microgrid, more verification simulations need to be completed. The future work is to integrate electric vehicles into the microgrid, and consider carbon emission constraints. At the same time, using machine learning or neural network and other advanced algorithms to more accurately predict renewable energy generation is also an important research direction, in order to further pursue the best benefit. On the other hand, the intraday market optimization stage is expected to be included in the future work.

**Author Contributions:** Conceptualization, J.D. and Y.W.; methodology, Y.W.; software, Y.W. and X.D.; validation, J.D., Y.W., and X.D.; formal analysis, Y.W.; investigation, Z.C.; resources, Y.Z.; data curation, Y.L.; writing—original draft preparation, Y.W.; writing—review and editing, Y.W. and Z.C.; visualization, X.D.; supervision, J.D.; project administration, J.D. All authors have read and agreed to the published version of the manuscript.

**Funding:** This research received no external funding.

**Institutional Review Board Statement:** Not applicable.

**Informed Consent Statement:** Not applicable.

**Data Availability Statement:** Not applicable.

**Conflicts of Interest:** The authors declare no conflict of interest.

## Appendix A

**Table A1.** Abbreviations and Acronyms.

| Abbreviations | Word |
| --- | --- |
| PV | photovoltaic |
| WT | wind turbine |
| GT | gas turbine |
| DR | demand response |
| IBDR | incentive-based demand response |
| PBDR | price-based demand response |
| MILP | mixed integer linear programming |
| DA | day-ahead |
| RT | real-time |
| LHS | Latin hypercube sampling |

**Table A2.** Relevant parameters for components in microgrid system.

| Time | Load (kW) | Day-Ahead Electricity Price/(USD/kWh) | Real-Time Electricity Price/(USD/kWh) |
|---|---|---|---|
| 1 | 657.817 | 0.0577 | 0.0768 |
| 2 | 640.751 | 0.0564 | 0.0737 |
| 3 | 615.284 | 0.0493 | 0.0729 |
| 4 | 499.450 | 0.0479 | 0.0710 |
| 5 | 519.889 | 0.0467 | 0.0673 |
| 6 | 530.361 | 0.0547 | 0.0726 |
| 7 | 708.096 | 0.0627 | 0.0752 |
| 8 | 792.761 | 0.0658 | 0.0768 |
| 9 | 867.985 | 0.0659 | 0.0774 |
| 10 | 870.208 | 0.0625 | 0.0789 |
| 11 | 910.671 | 0.0605 | 0.0760 |
| 12 | 940.382 | 0.0592 | 0.0731 |
| 13 | 920.851 | 0.0582 | 0.0732 |
| 14 | 900.046 | 0.0572 | 0.0666 |
| 15 | 849.933 | 0.0573 | 0.0700 |
| 16 | 751.563 | 0.0581 | 0.0708 |
| 17 | 881.136 | 0.0594 | 0.0757 |
| 18 | 952.000 | 0.0694 | 0.1018 |
| 19 | 1040.449 | 0.0863 | 0.1394 |
| 20 | 981.222 | 0.0942 | 0.1476 |
| 21 | 674.322 | 0.0895 | 0.1266 |
| 22 | 674.812 | 0.0816 | 0.0936 |
| 23 | 665.647 | 0.0801 | 0.0895 |
| 24 | 687.813 | 0.0613 | 0.0777 |

**Table A3.** Wind power output scenario.

| Time | 1 | 2 | 3 | 4 | 5 | 6 | 7 | 8 | 9 | 10 |
|---|---|---|---|---|---|---|---|---|---|---|
| 1 | 583.63 | 555.83 | 575 | 575 | 546.25 | 603.75 | 570.89 | 575 | 575 | 579.11 |
| 2 | 594.51 | 617.42 | 597.5 | 587.54 | 604.14 | 597.5 | 580.43 | 622.4 | 587.54 | 610.3 |
| 3 | 504.96 | 507.5 | 507.5 | 515.96 | 504.68 | 507.5 | 496.63 | 515.96 | 515.96 | 518.38 |
| 4 | 429.68 | 421.25 | 421.25 | 421.25 | 414.23 | 421.25 | 427.27 | 421.25 | 400.19 | 418.24 |
| 5 | 488.81 | 504.52 | 496.25 | 487.98 | 493.49 | 496.25 | 499.79 | 496.25 | 496.25 | 492.71 |
| 6 | 370.64 | 372.5 | 372.5 | 366.29 | 376.64 | 372.5 | 375.16 | 366.29 | 384.92 | 364.52 |
| 7 | 410.79 | 408.75 | 408.75 | 415.56 | 413.29 | 388.31 | 399.99 | 405.34 | 415.56 | 408.75 |
| 8 | 443.75 | 451.15 | 443.75 | 458.54 | 428.96 | 465.94 | 450.09 | 447.45 | 451.15 | 440.58 |
| 9 | 557.2 | 541.33 | 588 | 560 | 563.11 | 476 | 560 | 550.67 | 550.67 | 556 |
| 10 | 494.96 | 508.92 | 492.5 | 500.71 | 478.82 | 492.5 | 488.98 | 500.71 | 500.71 | 488.98 |
| 11 | 577.88 | 565.42 | 488.75 | 565.42 | 575 | 546.25 | 570.89 | 560.63 | 565.42 | 583.21 |
| 12 | 400.49 | 402.5 | 422.63 | 402.5 | 398.03 | 402.5 | 408.25 | 402.5 | 402.5 | 402.5 |
| 13 | 453.51 | 451.25 | 451.25 | 466.29 | 453.76 | 451.25 | 431.91 | 451.25 | 473.81 | 448.03 |
| 14 | 478.98 | 472.46 | 488.75 | 488.75 | 486.03 | 513.19 | 506.21 | 488.75 | 505.04 | 495.73 |
| 15 | 467.65 | 446.5 | 470 | 477.83 | 464.78 | 470 | 466.64 | 458.25 | 477.83 | 470 |
| 16 | 528.88 | 526.25 | 499.94 | 526.25 | 537.94 | 526.25 | 526.25 | 508.71 | 508.71 | 530.01 |
| 17 | 595.9 | 590 | 649 | 590 | 576.89 | 590 | 585.79 | 580.17 | 590 | 594.21 |
| 18 | 643.25 | 640 | 640 | 630.83 | 646.39 | 648 | 645.36 | 644.58 | 650 | 654.64 |
| 19 | 643.25 | 643.33 | 640 | 620 | 646.39 | 585 | 644.64 | 631.42 | 639.17 | 636.07 |
| 20 | 605.63 | 583.85 | 593.75 | 554.17 | 573.96 | 593.75 | 589.51 | 593.75 | 583.85 | 597.99 |
| 21 | 577.88 | 565.42 | 517.5 | 565.42 | 565.42 | 575 | 575 | 579.79 | 584.58 | 554.46 |
| 22 | 564.28 | 578.75 | 578.75 | 540.17 | 585.18 | 578.75 | 578.75 | 602.86 | 578.75 | 574.62 |
| 23 | 624.36 | 606.58 | 627.5 | 627.5 | 620.53 | 658.88 | 627.5 | 627.5 | 637.96 | 636.46 |
| 24 | 637 | 641.7 | 650 | 650 | 643.6 | 649.1 | 644.6 | 630.8 | 623.3 | 614.6 |

**Table A4.** PV power output scenario.

| Time | 1 | 2 | 3 | 4 | 5 |
|---|---|---|---|---|---|
| 1 | 0 | 0 | 0 | 0 | 0 |
| 2 | 0 | 0 | 0 | 0 | 0 |
| 3 | 0 | 0 | 0 | 0 | 0 |
| 4 | 0 | 0 | 0 | 0 | 0 |
| 5 | 0 | 0 | 0 | 0 | 0 |
| 6 | 0 | 0 | 0 | 0 | 0 |
| 7 | 24.23 | 24.23 | 24.23 | 24.59 | 23.02 |
| 8 | 88.3 | 85.5 | 87.69 | 87.25 | 89.15 |
| 9 | 150.67 | 154.05 | 156.6 | 151.25 | 152.78 |
| 10 | 172.18 | 165.21 | 170.98 | 172.14 | 167.13 |
| 11 | 233.38 | 234.26 | 183.17 | 248.85 | 230.61 |
| 12 | 295.16 | 264.27 | 297.13 | 299.42 | 289.99 |
| 13 | 213.42 | 205.78 | 219.42 | 207.14 | 175.09 |
| 14 | 173.06 | 176.04 | 179.12 | 153.23 | 194.53 |
| 15 | 144.36 | 140.84 | 130.64 | 145.94 | 140.84 |
| 16 | 101.5 | 105.48 | 98.88 | 102.18 | 102.18 |
| 17 | 56.61 | 56.24 | 59.54 | 56.14 | 56.71 |
| 18 | 7.64 | 7.59 | 7.46 | 7.73 | 7.78 |
| 19 | 0.09 | 0.09 | 0.09 | 0.09 | 0.08 |
| 20 | 0 | 0 | 0 | 0 | 0 |
| 21 | 0 | 0 | 0 | 0 | 0 |
| 22 | 0 | 0 | 0 | 0 | 0 |
| 23 | 0 | 0 | 0 | 0 | 0 |
| 24 | 0 | 0 | 0 | 0 | 0 |

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
