# Peer review of "Research on Decision Optimization Model of Microgrid Participating in Spot Market Transaction"

_sustainability, doi:10.3390/su13126577_

Round 1

Reviewer 1 Report

Review Comments on Manuscript entitled "Research on decision optimization model of microgrid participating in spot market transaction"(sustainability-1237890 ):

Please consider the following points during the revision of the manuscript:

The research article presents a thorough decision optimization model of microgrid participating in spot market transaction.

The theoretic background is solid and the article is enhanced with very interesting results depicted in numerous Figures.

The conclusions section sum up the research output while the readers can found a rich references list for a further study.

Author Response

Point 1: The research article presents a thorough decision optimization model of microgrid participating in spot market transaction.

Point 2: The theoretic background is solid and the article is enhanced with very interesting results depicted in numerous Figures.

Point 3: The conclusions section sum up the research output while the readers can found a rich references list for a further study.

Response: Thank you.

Reviewer 2 Report

The manuscript is well organized and offers all information required to replicate the results. The optimization model of microgrid participation in the day-ahead market transaction is sufficiently illustrated and documented. The results highlight the advantages and disadvantages of the various strategies tested.

There are, however, some detailed comments and questions the authors should consider and address in their revision, described below.

  1. I find however the discussion poor. The readers would benefit from a more insightful discussion of the results and a clear statement about the main conclusions drawn from the research carried out.
  2. Moreover, please use the English language for the axis title of Fig. 2.
  3. Clarify better the advantages of this approach. The authors must highlight the advantages of the proposed approach compared with the literature.
  4. I strongly consider that there are many limitations to the proposed method. Please emphasize them in a subsection.

Overall, this is a good paper, if the authors consider the points above.

Author Response

Point: The manuscript is well organized and offers all information required to replicate the results. The optimization model of microgrid participation in the day-ahead market transaction is sufficiently illustrated and documented. The results highlight the advantages and disadvantages of the various strategies tested.

Response: Thank you for your comments.

There are, however, some detailed comments and questions the authors should consider and address in their revision, described below.

Point 1: I find however the discussion poor. The readers would benefit from a more insightful discussion of the results and a clear statement about the main conclusions drawn from the research carried out.

Response 1: Thank you for your comment. We have added the 4.3 section of the discussion, please see the revisions using “Track Changes” functions in section 4.3.

Point 2: Moreover, please use the English language for the axis title of Fig. 2.

Response 2: Thank you for your comment. We have used the English language for the axis title of Fig. 2, please see the revisions using “Track Changes” functions in Figure 2.

Point 3: Clarify better the advantages of this approach. The authors must highlight the advantages of the proposed approach compared with the literature.

Response 3: Thank you for your comment. We have highlighted the advantages of the proposed approach compared with the literature, please see the revisions using “Track Changes” functions in section 1.2.

Point 4: I strongly consider that there are many limitations to the proposed method. Please emphasize them in a subsection.

Response 4: Thank you for your comment. We have added a description of the limitations of the proposed method in conclusion, please see the revisions using “Track Changes” functions in section 5.

Reviewer 3 Report

Comments and suggestions for Authors:

In the paper the authors are analysing the microgrid including wind power, photovoltaic (PV), gas turbine, battery storage and demand response with the scope to maximize the expected revenue of a microgrid in the spot market. The authors proposed a stochastic optimization method and a model to maximize the social-economic benefits of using an optimised grid, considering the uncertainty of wind and PV power. The Matlab model was not presented in the paper which makes it difficult to reproduce the analysis.

The paper provides an academic solution to the electricity generation problem. The main issue, in my opinion, is that the proposed solution is not economically practical for the microgrid. I suggest to the authors to choose only two systems to generate the electricity and redo the paper:

  • first system will cover the constant power (i.e. gas turbine)
  • second system will cover the load peak fluctuations (i.e. batteries).

Please see below several specific comments and questions:

  1. Line 402, 508, 509, 510 etc. – The entire paper needs to be checked for formatting, misspelling and grammar issues.
  2. The Introduction is lengthy and may be split in two sections Introduction and Literature Review to make it easier to read.
  3. I suggest using the USD currency in the paper for the cost analysis instead of the ¥ (yuan) currency for reader’s benefit.
  4. From the equation’s description are missing the units used, e.g. equation 4: V (wind speed) m/s, b (Weibull scale parameter) m/s.
  5. Line 274 – Calorific value of natural gas is 42-46 MJ/m3. In the article 36 MJ/m3 was used, please provide with a reference for this value.
  6. Line 505 – By solar intensity do you mean Solar Irradiance?
  7. Line 508, 509, 510 etc – It should be kWh instead of kW/h or kW∙h.
  8. Line 508 - The authors are confusing the electrical power with the electrical energy. What is the rated power of the battery?
  9. Line 510 - The rated power (50kW) of the gas turbine seems very low compared with the PV power (300kW) and wind turbine power (650kW).
  10. Figure 4 – The paper is lengthy with numerous abbreviations which is difficult to follow. I suggest avoiding using abbreviations where possible to ease the reading of the paper.
  11. Figure 4, 5 – It is not clear how the information in Figure 4 and 5 was calculated. A short description before the figures may help.
  12. Figure 5 – By electricity load do you mean electrical power? Also, based on the loading it seems that the chosen power ratings for PV and wind turbine (not taking into consideration the battery and gas turbine) are oversized. The generation system power rating should be optimised to reflect the load demand. An oversized system is not cost effective.
  13. Figure 9 – The figure presents the micro turbines power demand. The terminology used in the paper is not consistent. The authors should decide if they use turbines or micro turbines. How is the load demand covered between the time intervals 7h and 17h?
  14. It will be useful to show on a single figure how much load is covered by each generation system (PV, wind turbine, battery, gas turbine).
  15. The paper is missing a section describing the implementation of the model in Matlab.
  16. Please add a table, in the Case Study section, with all the input values used in the simulations conducted.
  17. Please add a table with the Abbreviations and Acronyms used in the paper.
  18. The conclusion section should be improved. Please provide specific conclusions from the study conducted.

Author Response

Point: In the paper the authors are analysing the microgrid including wind power, photovoltaic (PV), gas turbine, battery storage and demand response with the scope to maximize the expected revenue of a microgrid in the spot market. The authors proposed a stochastic optimization method and a model to maximize the social-economic benefits of using an optimised grid, considering the uncertainty of wind and PV power. The Matlab model was not presented in the paper which makes it difficult to reproduce the analysis.

The paper provides an academic solution to the electricity generation problem. The main issue, in my opinion, is that the proposed solution is not economically practical for the microgrid. I suggest to the authors to choose only two systems to generate the electricity and redo the paper:

first system will cover the constant power (i.e. gas turbine)

second system will cover the load peak fluctuations (i.e. batteries).

Response: Thank you for your comments.

Please see below several specific comments and questions:

Point 1: Line 402, 508, 509, 510 etc. – The entire paper needs to be checked for formatting, misspelling and grammar issues.

Response 1: Thank you for your comment. We have checked for formatting, misspelling and grammar issues, please see the revisions using “Track Changes” functions.

Point 2: The Introduction is lengthy and may be split in two sections Introduction and Literature Review to make it easier to read.

Response 2: Thank you for your comment. We have split “Introduction” in two sections, please see the revisions using “Track Changes” functions in section 1.

Point 3: I suggest using the USD currency in the paper for the cost analysis instead of the ¥ (yuan) currency for reader’s benefit.

Response 3: Thank you for your comment. We have used the USD currency in the paper for the cost analysis, please see the revisions using “Track Changes” functions.

Point 4: From the equation’s description are missing the units used, e.g. equation 4: V (wind speed) m/s, b (Weibull scale parameter) m/s.

Response 4: Thank you for your comment. We have increased the use of units in the paper, please see the revisions using “Track Changes” functions.

Point 5: Line 274 – Calorific value of natural gas is 42-46 MJ/m3. In the article 36 MJ/m3 was used, please provide with a reference for this value.

Response 5: Thank you for your comment. We have cited reference [34] in this paper to explain this question. The composition of natural gas and the proportion of methane (CH4) have a certain influence on the calorific value of natural gas, please see the revisions using “Track Changes” functions.

Point 6: Line 505 – By solar intensity do you mean Solar Irradiance?

Response 6: Thank you for your comment. We have changed the “solar intensity” to “solar Irradiance”, please see the revisions using “Track Changes” functions.

Point 7: Line 508, 509, 510 etc – It should be kWh instead of kW/h or kW∙h.

Response 7: Thank you for your comment. We have replaced the “kW/h” or “kW∙h” with “kWh”, please see the revisions using “Track Changes” functions in Table 2.

Point 8: Line 508 - The authors are confusing the electrical power with the electrical energy. What is the rated power of the battery?

Response 8: Thank you for your comment. We have distinguished “electrical power” and “electrical energy”, the rated charging power of the battery is 15kW, and the rated discharging power of the battery is 20kW, please see the revisions using “Track Changes” functions.

Point 9: Line 510 - The rated power (50kW) of the gas turbine seems very low compared with the PV power (300kW) and wind turbine power (650kW).

Response 9: Thank you for your comment. We have redone the microgrid system and the rated power of the gas turbine has been modified to 100kW, please see the revisions using “Track Changes” functions.

Point 10: Figure 4 – The paper is lengthy with numerous abbreviations which is difficult to follow. I suggest avoiding using abbreviations where possible to ease the reading of the paper.

Response 10: Thank you for your comment. We have added a table with the abbreviations in the paper, please see the revisions using “Track Changes” functions.

Point 11: Figure 4, 5 – It is not clear how the information in Figure 4 and 5 was calculated. A short description before the figures may help.

Response 11: Thank you for your comment. The load curve is obtained based on the user's historical load curve, and the market electricity price data is obtained based on fuzzy clustering., please see the revisions using “Track Changes” functions.

Point 12: Figure 5 – By electricity load do you mean electrical power? Also, based on the loading it seems that the chosen power ratings for PV and wind turbine (not taking into consideration the battery and gas turbine) are oversized. The generation system power rating should be optimized to reflect the load demand. An oversized system is not cost effective.

Response 12: Thank you for your comment. “Electricity load” refers to “electrical power”, and we have modified “electricity load” to “load”. In addition, we have redone the microgrid system to reflect the load demand, please see the revisions using “Track Changes” functions.

Point 13: Figure 9 – The figure presents the micro turbines power demand. The terminology used in the paper is not consistent. The authors should decide if they use turbines or micro turbines. How is the load demand covered between the time intervals 7h and 17h?

Response 13: Thank you for your comment. We have decided to use gas turbines, and between 7h and 17h, the load demand is covered by renewable energy output power, gas turbines, battery storage, demand response, and market transaction electricity, please see the revisions using “Track Changes” functions in Figure 8.

Point 14: It will be useful to show on a single figure how much load is covered by each generation system (PV, wind turbine, battery, gas turbine).

Response 14: Thank you for your comment. We have added a figure of the output power of microgrid system components, please see the revisions using “Track Changes” functions in Figure 8.

Point 15: The paper is missing a section describing the implementation of the model in Matlab.

Response 15: Thank you for your comment. We have added a description of the implementation of the model in Matlab. please see the revisions using “Track Changes” functions in 3.3 section.

Point 16: Please add a table, in the Case Study section, with all the input values used in the simulations conducted.

Response 16: Thank you for your comment. We have added a table with all the input values, please see the revisions using “Track Changes” functions in Table2, TableA2, TableA3 and TableA4.

Point 17: Please add a table with the Abbreviations and Acronyms used in the paper.

Response 17: Thank you for your comment. We have added a table with the Abbreviations and Acronyms used in the paper, please see the revisions using “Track Changes” functions in Table A1.

Point 18: The conclusion section should be improved. Please provide specific conclusions from the study conducted.

Response 18: Thank you for your comment. We have revised the conclusion section, please see the revisions using “Track Changes” functions.

Ends.

Round 2

Reviewer 3 Report

Comments and suggestions for Authors:

I would like to thank authors for the detailed response. My comments were addressed which improved the paper.